# TF/PAR2 Signaling Axis Supports the Protumor Effect of Neutrophil Extracellular Traps (NETs) on Human Breast Cancer Cells

**DOI:** 10.3390/cancers16010005

**Published:** 2023-12-19

**Authors:** Karina Martins-Cardoso, Aquiles Maçao, Juliana L. Souza, Alexander G. Silva, Sandra König, Remy Martins-Gonçalves, Eugenio D. Hottz, Araci M. R. Rondon, Henri H. Versteeg, Patrícia T. Bozza, Vitor H. Almeida, Robson Q. Monteiro

**Affiliations:** 1Institute of Medical Biochemistry Leopoldo de Meis, Federal University of Rio de Janeiro, Rio de Janeiro 21941-902, Brazil; kmcardoso@bioqmed.ufrj.br (K.M.-C.); aquiles.junior@bioqmed.ufrj.br (A.M.); juliana.souza@bioqmed.ufrj.br (J.L.S.); alexander.silva@bioqmed.ufrj.br (A.G.S.); vhluna@bioqmed.ufrj.br (V.H.A.); 2Institute of Biomedical Sciences, Federal University of Rio de Janeiro, Rio de Janeiro 21941-902, Brazil; sandrakon@gmail.com; 3Laboratory of Immunopharmacology, Oswaldo Cruz Institute, Oswaldo Cruz Foundation, Rio de Janeiro 21040-360, Brazil; remymgoncalves@gmail.com (R.M.-G.); pbozza@ioc.fiocruz.br (P.T.B.); 4Laboratory of Immunothrombosis, Department of Biochemistry, Federal University of Juiz de Fora (UFJF), Rio de Janeiro 23890-000, Brazil; eugenio.hottz@ufjf.br; 5Einthoven Laboratory for Experimental Vascular Medicine, Department of Thrombosis and Hemostasis, Leiden University Medical Center, 333 ZA Leiden, The Netherlands; a.m.da_rocha_rondon@lumc.nl (A.M.R.R.); h.h.versteeg@lumc.nl (H.H.V.)

**Keywords:** neutrophil extracellular traps, tissue factor, protease-activated receptor 2, breast cancer, inflammatory cytokines

## Abstract

**Simple Summary:**

Neutrophil extracellular traps (NETs) contribute to tumor progression at different stages, such as primary growth, metastasis, angiogenesis, and cancer-associated thrombosis. The knowledge of biomolecular mechanisms of this process is crucial to developing strategies to mitigate tumor progression, mainly, metastasis. NETs promote the activation of proinflammatory pathways and trigger the epithelial-mesenchymal transition (EMT) process. In this work, we demonstrate that NETs enhance the expression of tissue factor (TF) in breast tumor cells, thus increasing the procoagulant activity. NETs also promote protease-activated receptor 2 (PAR2) signaling, leading to the expression of pro-tumor cytokines and factors associated with EMT. These phenomena are supported by in silico gene correlation analysis of TF/PAR2 and pro-tumor genes analyzed in samples from breast cancer patients. Our results suggest that the TF/PAR2 signaling axis contributes to the pro-cancer effects of NETs in human breast cancer cells.

**Abstract:**

Neutrophil extracellular traps (NETs) have been implicated in several hallmarks of cancer. Among the protumor effects, NETs promote epithelial-mesenchymal transition (EMT) in different cancer models. EMT has been linked to an enhanced expression of the clotting-initiating protein, tissue factor (TF), thus favoring the metastatic potential. TF may also exert protumor effects by facilitating the activation of protease-activated receptor 2 (PAR2). Herein, we evaluated whether NETs could induce TF expression in breast cancer cells and further promote procoagulant and intracellular signaling effects via the TF/PAR2 axis. T-47D and MCF7 cell lines were treated with isolated NETs, and samples were obtained for real-time PCR, flow cytometry, Western blotting, and plasma coagulation assays. In silico analyses were performed employing RNA-seq data from breast cancer patients deposited in The Cancer Genome Atlas (TCGA) database. A positive correlation was observed between neutrophil/NETs gene signatures and TF gene expression. Neutrophils/NETs gene signatures and PAR2 gene expression also showed a significant positive correlation in the bioinformatics model. In vitro analysis showed that treatment with NETs upregulated TF gene and protein expression in breast cancer cell lines. The inhibition of ERK/JNK reduced the TF gene expression induced by NETs. Remarkably, the pharmacological or genetic inhibition of the TF/PAR2 signaling axis attenuated the NETs-induced expression of several protumor genes. Also, treatment of NETs with a neutrophil elastase inhibitor reduced the expression of metastasis-related genes. Our results suggest that the TF/PAR2 signaling axis contributes to the pro-cancer effects of NETs in human breast cancer cells.

## 1. Introduction

Tissue factor (TF) is a 47 KDa transmembrane glycoprotein with a well-established role in activating the hemostatic system. TF is constitutively expressed in subendothelial cells, and after the rupture of a blood vessel, it is exposed and interacts with factor VII, forming the binary complex TF/FVIIa. Alternatively, TF may be expressed in a variety of cell types upon specific activation contexts. TF/FVIIa complex activates the extrinsic blood coagulation cascade, producing fibrin and promoting clot formation [1]. TF also possesses a non-coagulant function that results in intracellular signaling [2]. The TF/FVIIa complex binds to coagulation factor X, forming the ternary structure TF/FVIIa/Xa that can cleave and activate protease-activated receptors (PARs) [2,3,4]. The signaling mediated by the interaction between TF/PAR2 leads to the activation of MAPK and PI3K signaling pathways that regulate numerous cellular processes, including the ones leading to malignancy [5,6,7]. Increased expression of TF is associated with cancer progression [8,9,10]. Moreover, the TF/PAR2 signaling axis plays a role in angiogenesis, tumor growth, cell motility, cell survival, and the production of proinflammatory molecules [6,11,12,13].

Epithelial–mesenchymal transition (EMT) is a complex cellular program that regulates a set of protumor genes related to tumor cell migration, invasion, and metastatic properties [14,15]. A correlation between EMT and TF expression has been proposed [16,17,18]. Activation of EMT in breast carcinoma cell lines leads to the augmentation of TF expression, a phenomenon reversed by the silencing of ZEB1 (Zinc finger E-box-binding homeobox 1), a master regulatory factor of the EMT [18]. More recently, it was observed that the transmembrane glycoprotein CD44, a cancer stem cell (CSC) marker, which is also associated with EMT, regulates the expression of TF in breast cancer cells, resulting in enhanced tumor cell procoagulant activity and metastatic dissemination [19]. TF can also induce EMT, corroborating the evidence that there is a linkage between them [20]. Blocking of the TF signaling with a monoclonal anti-human TF antibody reduced EMT and CSC programs in breast cancer cells, affecting cell invasion and spheroid formation [20].

Several lines of evidence implicate neutrophil extracellular traps (NETs) in tumor progression [21,22]. Composed of double-stranded nuclear DNA decorated with granular/nuclear/cytoplasmic proteins, NETs were first described as a host defense mechanism against pathogens [23]. Brinkman and collaborators showed that the DNA web captures the pathogenic microorganisms while the proteins exert a cytotoxic effect [23]. NET formation was further associated with the progression of numerous non-infectious diseases, ranging from autoimmunity to thrombosis [24]. In the tumor context, NETs participate in several stages of tumor establishment and metastasis dissemination [21,22]. Studies have shown that NETs can improve primary tumor growth [25,26], enhance cell migration [27], sequester circulating tumor cells, and support metastasis [28,29]. In addition to promoting a tumor-associated inflammatory response [30] and resistance to radiotherapy [31] and chemotherapy [32], NETs have also been involved with cancer-associated thrombosis [33,34]. Furthermore, NET components can interact with different tumor cell receptors, activating cell signaling pathways that mediate various protumor responses [21,22,35,36].

We have previously demonstrated that NETs may promote pro-metastatic features in human breast carcinoma cells through the activation of the EMT process [37]. NETs induced the gene expression of ZEB1, CD44, and other protumoral and proinflammatory factors. In this context, this study aimed to evaluate whether NETs can induce TF expression in human breast carcinoma cells. Treatment of MCF7 and T-47D cells with isolated NETs promoted TF expression and enhanced the procoagulant activity of tumor cells. Moreover, it was observed that the TF/PAR2 axis supports NETs-induced gene expression of protumor factors but does not regulate the activation of EMT. Together, our results indicate that NETs enhance the procoagulant properties of breast cancer cells as well as their protumor features, at least in part, through the TF/PAR2 signaling pathway.

## 2. Materials and Methods

### 2.1. Cell Culture and Reagents

The experiments were conducted in vitro using the breast cancer cell lines MCF7, T-47D, and MDA-MB-231 obtained from the Rio de Janeiro Cell Bank (Rio de Janeiro, RJ, Brazil). TF-knockout MDA-MB-231 cells and wild-type MDA-MB-231 cells were previously obtained and characterized [38]. Maintenance of the cells was performed utilizing DMEM (Dulbecco’s Modified Eagle Medium, Thermo Fisher Scientific, Waltham, MA, USA) with the addition of 10% fetal bovine serum–FBS (Cultilab, Campinas, Brazil) and 1% penicillin/streptomycin (Thermo Fisher Scientific), which was incubated at 37 °C in 5% CO_2_ atmosphere. Seeded cells were starved for at least 8 h before incubation with NETs to perform all experiments. Then, starved cells were treated with isolated NETs (500 ng/mL) for 24 h. Cells were further washed twice with phosphate-buffered saline (PBS) and used in the assays. A monoclonal anti-human TF antibody, 10H10, was kindly provided by Wolfram Ruf (Johannes Gutenberg University Medical Center, Mainz, Germany; and Department of Immunology and Microbiology, The Scripps Research Institute, La Jolla, CA, USA). Phorbol 12-myristate 13-acetate (PMA), 2-(2-Amino-3-methoxyphenyl)-4H-1-benzopyran-4-one (PD98059), 1,9-Pyrazoloanthrone, Anthrapyrazolone (SP600125), 4-(4-Fluorophenyl)-2-(4-methylsulfinylphenyl)-5-(4-pyridyl)-1H-imidazole (SB203580), and 2-6-Bromo-1,3-benzodioxol-5-yl)-N-(4-cyanophenyl)-1-[(1S)-1-cyclohexylethyl]-1H-benzimidazole-5-carboxamide (Az3451) were purchased from Merck (Darmstadt, Germany).

### 2.2. Gene Expression Correlation Analysis

For bioinformatics analysis, a transcriptome database containing 1100 breast cancer samples available at The Cancer Genome Atlas (TCGA-BRCA) was accessed using the GEPIA2 online platform—http://gepia2.cancer-pku.cn/#index (accessed on 10 October 2022) [39]. Herein, we analyzed the Spearman rank correlation between the gene expression of TF (*F3*) or PAR2 (*F2RL1*) and a set of neutrophil-related genes previously defined [40] or two distinct neutrophil extracellular traps gene signatures [41,42].

### 2.3. Neutrophil Isolation and NETs Obtention

Fresh blood from healthy donors was collected using sodium citrate (3.8%) as an anticoagulant, at a 1:9 *v*/*v* proportion. Neutrophils were purified through density gradient centrifugation using Histopaque-1077 (Merck, Darmstadt city, Germany). Afterward, neutrophils were stimulated with PMA (500 nM) for 3 h. NETs were isolated according to the simplified protocol described by Najmeh et al., 2015 [43]. As seen in the Appendix A, DNA staining with DAPI correlates with citrullinated Histone 3 staining in our NET preparations. Isolated NETs were kept at 4 °C for no more than 24 h. These procedures were approved by an institutional ethical committee (register 82933518.0.0000.525) from Clementino Fraga Filho University Hospital, Federal University of Rio de Janeiro.

### 2.4. Quantitative RT-PCR (qRT-PCR)

The total RNA of 5 × 10^5^ cells was extracted using TRIzol Reagent, and 1 µg of RNA per sample was purified with DNase I at 65 °C for 10 min. Then, reverse transcription PCR was performed using a high-capacity cDNA Reverse Transcription Kit (Applied Biosystems, Foster City, CA, USA). Real-time polymerase chain reaction (PCR) was assessed to amplify the complementary DNA (cDNA) using the SYBR Green Real-Time PCR Master Mix on the StepOnePlus Real-Time PCR System (both from Thermo Fisher Scientific). Appendix A shows the primer sequences of previously tested primers with reaction efficiencies in the range of 90–110%. Gene expression was normalized using *GAPDH* as the reference gene. To analyze the relative fold change, we employed the 2^−ΔΔCT^ method.

### 2.5. Flow Cytometry Analysis

An amount of 1 × 10^6^ cells/mL suspended in serum-free DMEM medium was washed twice with cold FACS buffer (PBS containing 0.01% sodium azide and 3% FBS) and labeled with PE-conjugated antibody anti-CD142 (TF) (BD Pharmingen, EUA) for 30 min at room temperature and fixed with 4% paraformaldehyde. The supernatant was discarded, and cells were resuspended in HT buffer (10 mM N-2-hydroxyethyl piperazine-N′-2-ethane sulfonic acid (HEPES), 137 mM NaCl, 2.8 mM KCl, 1 mM MgCl_2_, 6H_2_O, 12 mM NaHCO_3_, 0.4 mM Na_2_HPO_4_, 5.5 mM glucose, 0.35% bovine serum albumin (pH 7.4)). Isotype immunoglobulin G (IgG) conjugated with the same fluorochrome was used as the negative control. A flow cytometer (BD FACSCalibur, Becton, Dickinson and Company, Franklin Lakes, NJ, USA) was used, and data were further analyzed using FlowJo software (Version 10).

### 2.6. Plasma Clotting Assays

The procoagulant activity of tumor cells was evaluated by recalcification assay using human platelet-poor plasma (PPP), as previously described [44]. Blood was collected as described above, and PPP was obtained upon centrifugation at 1000× *g* for 10 min. Tumor cells (1 × 10^5^) in 50 µL of PBS were incubated with 50 µL of human PPP. After 1 min incubation at 37 °C, 100 μL of 25 mM CaCl_2_ was added, and clotting times were monitored on an Amelung KC-4 Coagulation analyzer (Grifols Diagnostic Solutions Inc., Emeryville, CA, USA).

### 2.7. Western Blot

Starved cells (1 × 10^6^ cells/sample) were cocultured with isolated NETs (500 ng/mL) in serum-free DMEM medium for 24 h. After the treatment, cells were lysed, and proteins were quantified using the Lowry method (DC protein assay, Bio-Rad, Hercules, CA, USA). Protein lysates (30–50 μg) were run on 6–10% SDS–PAGE and transferred onto PVDF membrane, es (GE Healthcare, São Paulo, SP, Brazil). Membranes were blocked and incubated overnight at 4 °C, with the following primary antibodies against phosphor-ERK (1:1000; #9101; Cell Signaling Technology, Danvers, MA, USA), ERK (1:1000; #9102; Cell Signaling Technology, Danvers, MA, USA), β-actin (1:1000; #8457; Cell Signaling Technology, Danvers, MA, USA), fibronectin (1:750; #F3648; Merck, Darmstadt, Germany), and E-cadherin (1:10,000; #610182; BD Biosciences, Franklin Lakes, NJ, USA). Afterward, HRP-conjugated secondary antibodies (DakoCytomation, Glostrup, Denmark) were added at room temperature for 1 h. Immunoblots were detected using a chemiluminescence substrate, Clarity Western ECL Substrate (Bio-Rad, Hercules, CA, USA).

### 2.8. ELISA

Supernatants from 5 × 10^5^ cells cultured in the absence or presence of NETs in a 6-well plate were quantified by enzyme-linked immunosorbent assay (ELISA) using a commercial kit for IL-8 (PeproTech, Inc, Cranbury, NJ, USA) following the manufacturer’s protocol.

### 2.9. Statistical Analysis

GraphPad Prism 5 (GraphPad Prism 5 Software, San Diego, CA, USA) was applied for statistical analysis. Data are shown in bar graphs representing the mean and standard deviation. To determine the statistical significance of the treatment with NETs in the qPCR, ELISA, and migration assays, the unpaired *t*-test was performed. One-way ANOVA using Tukey post-test was performed in the qPCR, and Western blotting analysis comparing cells treated with NETs in the presence or absence of inhibitors of MEK, p38, JNK, TF, PAR2, and NE. The non-parametric Spearman test was used to correlate the RNA-seq values (FPKM). Statistical significance was considered if the *p*-value was ≤ 0.05.

## 3. Results

### 3.1. NET Signatures Are Positively Correlated with Gene Expression of TF and PAR2 in Breast Cancer Patients

In a previous study, we observed significant correlations between EMT-related genes and neutrophil-related genes using in silico analysis [37]. Considering that TF expression has been associated with the EMT process [16,17,18,19], we once more assessed the TCGA mRNA database of 1100 breast cancer patients to interrogate whether neutrophil/NET gene signatures correlate with TF or PAR2 mRNA expression. As seen in Figure 1, both TF and PAR2 were positively correlated with an eight-gene neutrophil signature [40]. We further employed two distinct proposed NET signatures for comparative analysis. The first NET signature, composed of six genes, was used in a head and neck squamous cell carcinoma study [41]. The second proposed NET signature is composed of 23 genes, and it has been used in multiple cancer types from various databases [42]. Through the GEPIA2 online platform (available at http://gepia2.cancer-pku.cn/#index, accessed on 10 October 2022), Spearman rank was performed.

The analyses indicate that both the neutrophil-gene signature and the NET-gene signatures presented a significant positive correlation with TF (*F3*) and PAR2 (*F2RL1*) expression, with a Rank correlation in the range of 0.17 to 0.37 (Figure 1). These analyses demonstrate a possible association between the presence of NETs in the malignant tissue and the expression of TF and PAR2 in breast tumors.

### 3.2. NETs Induce TF Expression and Enhance the Procoagulant Activity in Tumor Cell Lines

MCF7 and T-47D cells, which poorly expresses TF, were treated for 24 h with isolated NETs. Further, TF expression was evaluated by the qRT-PCR approach. We observed a significant increase in the TF gene expression in both cell lines (Figure 2A,D). Also, we detected a higher TF protein labeling in both MCF7 and T-47D cells treated with NETs, as assessed by flow cytometric analysis (Figure 2B,E). In accordance with qRT-PCR and FACS analyses, NET-treated cells accelerated the plasma coagulation. Together, these results suggest that NETs increase TF expression, thus altering the coagulant activity of cultured breast cancer cells.

### 3.3. MAPK Signaling Pathways Regulate NET-Induced TF Expression

The mitogen-activated protein kinase (MAPK) cascades are intracellular signal transduction pathways involved in the cell response to mitogens and stress-related stimuli. Extracellular signal-regulated kinase 1 and 2 (ERK1/2), p38, and c-Jun N-terminal kinase (JNK) have been consistently described as MAPK cascades regulated in the tumor progression [45,46]. Firstly, we evaluated the ERK phosphorylation (p-ERK) generated by NET treatment in MCF7 cells (Figure 3A). We observed an increase in the p-ERK levels (approximately 25%) in NET-treated cells compared to non-stimulated cells. We employed commercial pharmacological inhibitors to evaluate the possible involvement of MAPK pathways in NET-induced TF expression. PD98059, an MEK inhibitor, strongly suppressed the NET-driven TF upregulation (Figure 3B). MCF7 cells treated with NETs in the presence of SP600125 (JNK inhibitor) also showed an attenuated TF gene expression (Figure 3C). However, treatment with SB03580, a p38 inhibitor, did not impair the effect of NETs on the TF gene expression (Figure 3D). Therefore, the results suggest that NETs modulate TF expression through the activation of ERK/JNK MAP kinases.

### 3.4. TF Signaling Contributes to the Protumor Effects of NETs

Beyond the procoagulant effect, TF can promote various protumor responses through cell signaling, mediated or not, by the PAR2 receptor [2,3,4,7]. To assess a possible role of TF signaling in breast cancer cells treated with NETs, we employed the monoclonal antibody, 10H10, which blocks the TF/PAR2 signaling without significant effect on TF procoagulant activity [11]. As seen in Figure 4, treatment with 10H10 prior to exposure to NETs strongly prevents the gene expression of the proinflammatory cytokines, interleukin 8 (*CXCL8*) and interleukin 6 (*IL6*), in MCF7 and T-47D cells. In MCF7, but not in T-47D cells, 10H10 attenuated the NET-induced expression of *CD44*, a marker of tumor stemness. In contrast, 10H10 decreased the gene expression of *ZEB1*, an EMT-related transcription factor, in T-47D, but not in MCF7 cells.

To further explore the role of TF signaling in NET-mediated effects, we next employed TF-knockout (KO) MDA-MB-231 cells. As seen in Figure 5, the exposure of wild-type (WT) MDA-MB-231 to NETs, for 24 h, promoted an increased expression of *CXCL8*, *IL1B*, *IL6*, *MMP9* (metalloproteinase 9), and *PTGS2* (cyclooxygenase 2) genes. In contrast, most of these genes have not been modulated by NETs in KO MDA-MB-231 cells. Accordingly, high levels of secreted IL-8 were observed in WT MDA-MB-231 cells, but not in KO MDA-MB-231 cells treated with NETs (Figure 5B). Interestingly, blocking TF expression in the MDA-MB-231 cell was not sufficient to prevent the effect of NETs on cell migration (Appendix A).

### 3.5. PAR2 Contributes to NET-Mediated Protumor Effects

Activation of PAR2 in the tumor microenvironment may be elicited by various proteases, including TF/FVIIa complex, therefore generating intracellular responses that have been linked to proliferation, cell migration, cytokine production, angiogenesis, and other protumor responses [2,3,47]. Therefore, we next evaluated the possible contribution of PAR2 activation to the effects of NETs on tumor cells. A commercial PAR2 antagonist, Az3451 [48], abrogated the NET-induced expression of TF in both cell lines (Figure 6A,B). A similar inhibitory profile was observed in the *CD44* gene expression, which was partially impaired in MCF7 cells and entirely abolished in T-47D cells. Interestingly, *CXCL8* gene expression was not affected by the presence of PAR2 antagonist in MCF7 cells but significantly inhibited in T-47D cells treated with NETs. A similar pattern was observed for *ZEB1* gene transcription. Our results point to PAR2 as a mediator of some of the NET protumor responses in breast cancer cells.

### 3.6. TF and PAR2 Play a Minor Role in NET-Induced EMT

Previously, Bourcy and colleagues demonstrated that EMT activation could regulate the TF expression and the procoagulant activity in the breast cancer cell line, MDA-MB-468 [18]. We sought to evaluate whether the TF induced by NETs can regulate the EMT activation as a positive feedback loop. After 24 h of treatment with NETs, we assessed the protein expression of EMT markers, and, as expected, MCF7 cells increased the fibronectin protein level and decreased E-cadherin expression (Figure 7A,B). However, we were unable to detect any effect of the 10H10 antibody or the PAR2 antagonist, Az3451, on both EMT markers modulated by NETs in this cell line. We have also investigated the capacity of NETs to induce EMT in T-47D cells. As seen in Figure 7C,D, we failed to detect minor effects of NETs on both EMT markers in T-47D cells.

### 3.7. Neutrophil Elastase Present in NETs Can Modulate the Protumor Gene Expression

Finally, we sought to assess whether the effect of NETs on the modulation of protumor genes occurs through the action of neutrophil elastase (NE). NE is one of the main components present in NETs and, according to previous evidence, a potential PAR2 activator [49,50]. Indeed, the treatment of tumor cells with a commercial NE inhibitor (NEi) changed the NET-induced gene expression profile in both cell lines. In MCF7 cells, the NEi reduced the effect of NETs in *CD44*, *IL6*, and *F3* gene expression, but not *ZEB1* and *CXCL8* (Figure 8A). In contrast, NEi impaired the NET-induced expression of almost all genes (except for *ZEB1*) in T-47D cells (Figure 8B). In conclusion, these results demonstrate that NE present in NETs can regulate the protumor gene expression in cultured breast cancer cells.

## 4. Discussion

Over the last decade, the role of NETs in tumor biology has been extensively investigated. In this context, the participation of NETs in tumor biology is well established [21,22]. Recent studies have supported a role for NETs in tumor progression, not only as a physical structure for sequestering circulating cancer cells but also as an inducer of the intracellular response in tumor cells [21,35,36]. We previously provided evidence that NETs activate the EMT program in human breast cancer cells and promote the acquisition of a prometastatic profile [37]. Consistent with our data, in vitro experiments demonstrated that NETs promote EMT activation in pancreatic, gastric, and colon cancer cells [51,52,53]. NET inhibitors blocked EMT activation and reduced metastasis in a murine breast cancer model [32].

Previous studies have demonstrated that EMT drives TF expression in vitro in different carcinoma models [16,17,18,19]. In a murine model, TF or ZEB1 silencing impaired the metastatic niches in the lungs [18]. Accordingly, Sun and co-workers showed the regulation of TF expression by the miR200a/ZEB1 axis in glioma cells [54]. In addition, Villard and collaborators showed that CD44, a CSC marker, controls the TF expression in the MDA-MB-468 breast cancer cell line, and, consequently, its procoagulant activity [19]. Therefore, EMT has been linked to an increase in TF-dependent procoagulant properties and tumor progression. Considering our previous findings that NETs increase gene expression of *ZEB1* and *CD44* in MCF7 breast cancer cells [37], we hypothesized that NETs could induce EMT activation through TF upregulation. In fact, here we demonstrated that exposure of cultured breast cancer cell lines to isolated NETs increases TF expression, thus enhancing their procoagulant activity.

A number of studies have demonstrated that NETs activate the MAPK pathways in different tumor cell types, including diffuse large B-cell lymphoma, colorectal, and gastric cancer cells [55,56,57]. MAPKs are key signal transducers that regulate a variety of processes, including cell proliferation, apoptosis, cell differentiation, and stress response [58]. In mammals, four different MAPK cascades have been identified: ERK1/2, JNK, p38, and ERK5. The ERK5 cascade is less studied and less understood than the others [58]. Interestingly, the JNK and the ERK1/2 signaling pathways induce TF gene expression in glioblastoma cell lines by enhancing AP-1 transcriptional activity [59]. Indeed, the basal human TF promoter (−250 to +1) contains two AP-1 binding sites [59]. Here, we employed commercial inhibitors to investigate the role of each MAPK pathway in the response mediated by NETs. PD98059 inhibitor is highly selective against MEK, upstream of ERK1/2, and SP600125 is a specific inhibitor of JNK. SB203580 inhibits specifically the p38 catalytic activity. We observed the contribution of ERK and JNK MAP kinases, but not p38, in the TF gene regulation induced by NETs.

A correlation between TF expression levels and tumor cell aggressiveness has been consistently demonstrated in breast cancer cell lines [60,61]. Further, in breast cancer patients, TF expression was shown to be increased in both primary tumor and plasma samples compared to normal controls, pointing to an important role in breast cancer progression [8]. Ünlü and colleagues recently demonstrated that TF inhibition in breast cancer cells reduces EMT activation [20]. In this context, the blockade of TF signaling with a monoclonal antibody impaired EMT activation in vitro and reduced metastasis in vivo [20].

In addition to TF, PAR2 has been also linked to EMT programs in cancer cells [62,63]. PAR2 receptor is activated by proteases that can cleave at different sites of its extracellular NH_2_-terminal. Depending on the cleavage site, distinct signals are evoked, and the intracellular responses are distinct (biased agonism). Studies indicate that NE cleaves PAR2 at the Ser68–Val69 site, which activates the MAPK signaling pathway, unlike cathepsin G, which cleaves the Phe65–Ser66 site and does not activate the MAPK pathway [49,50]. Indeed, neutrophil elastase is a major serine protease associated with NETs [64]. In the tumor context, NE has been implicated in tumorigenesis, primary tumor growth, and establishment of metastasis, regulating the EMT program in pancreatic tumor cells [65,66]. Albrengues and colleagues demonstrated that NE and MMP9, both present in NETs, cleave laminin, inducing the proliferation of dormant cancer cells [67]. In colorectal cancer cells treated with NETs, NE activated TLR4, promoting tumor growth [25]. Here, we demonstrate that NE is one of the possible effectors of NETs in regulating the expression of TF and other protumor genes. Interestingly, all genes downregulated by Az3451 (PAR2 inhibitor) were also modulated by NEi. We suggest that NE can act, at least in part, through PAR2 cleavage/activation. However, further studies are needed to confirm this hypothesis. Remarkably, the inhibition of either TF or PAR2 signaling was insufficient to prevent cell migration (Appendix A) and EMT activation. A possible role of the TF/PAR2 signaling axis for EMT induced by NETs will require more investigation. A possible explanation for this observation relies on the several NET components that may elicit TF/PAR2-independent cell signaling pathways [21]. For example, in pancreatic cancer, NETs promoted cell migration, invasion, and subsequent EMT activation via the EGFR/ERK pathway [27]. Other receptors have also been identified as mediators between NETs and protumor effects, such as RAGE [36], toll-like receptors (TLRs) [25,30,57], and integrins [68].

In summary, our findings indicate that components of NETs activate ERK/JNK pathways resulting in increased expression of TF and procoagulant activity. Furthermore, NETs stimulate gene expression of protumor factors (such as IL1β, IL6, and CXCL8) through the TF/PAR2 axis (Figure 9).

The understanding is that NETs are not just scaffolds but are also structures capable of triggering intracellular responses, and modulating the behavior of cancer cells is fundamental in tumor biology. The elucidation of protumor NET components, their tumor cell receptors, and major signaling pathways may pave the way for the development of novel therapeutic strategies.

## 5. Conclusions

Our findings demonstrate that NET-induced TF expression plays a role in regulating the expression of protumor genes and procoagulant activity in human breast carcinoma cells.

## Figures and Tables

**Figure 1 cancers-16-00005-f001:**
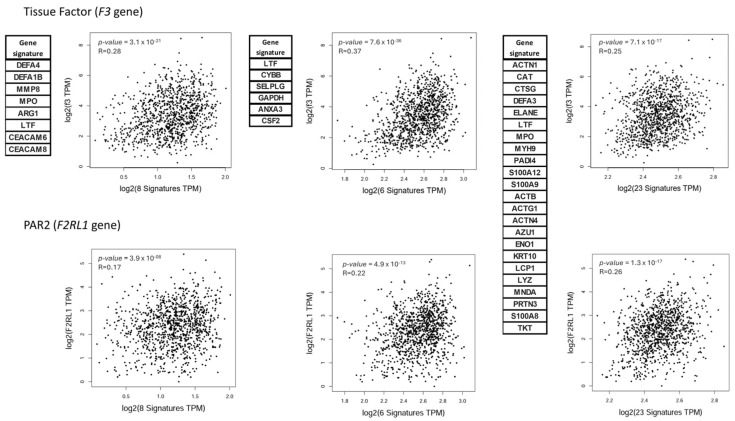
A NET gene signature is positively correlated with the gene expression of TF and PAR2 in breast cancer samples. Spearman’s correlation analysis between TF (*F3*) or PAR2 (*F2RL1*) and neutrophil/NET gene signatures. Neutrophil-related gene signature (**left**) based on systemic lupus erythematosus, NET gene signature validated in head and neck squamous cell cancer (**middle**), and NET gene signature for pan-cancer prognosis (**right**). RNAseq data based on 1085 breast cancer patient samples deposited at TCGA. R = rank correlation. *p*-value < 0.001 is statistically significant.

**Figure 2 cancers-16-00005-f002:**
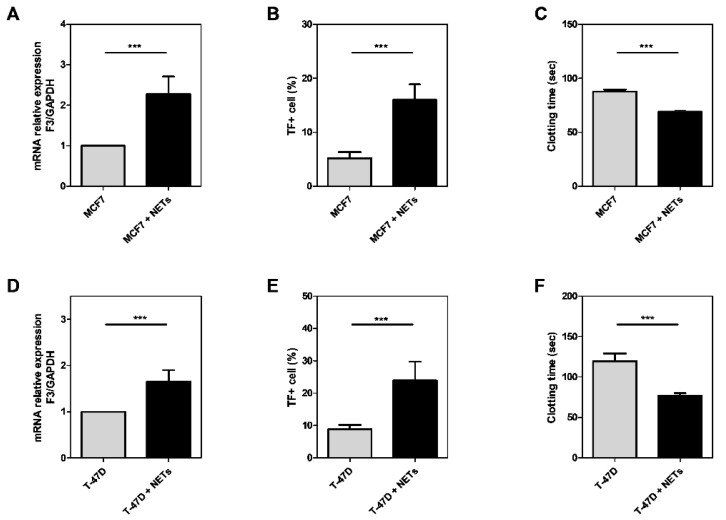
NETs increase TF expression and procoagulant activity in breast cancer cell lines. MCF7 and T-47D cells were starved and stimulated with NETs (500 ng/mL) for 24 h. TF mRNA expression was evaluated by qRT-PCR, and *GAPDH* was used as a reference gene. The relative expression of mRNA was calculated using the ΔΔCT method (**A**,**D**). Flow cytometry was performed using PE-conjugated antibody anti-CD142 (**B**,**E**). A clotting assay was carried out using platelet-poor plasma incubated with breast cancer cells. The reaction was initiated with CaCl_2_ (**C**,**F**). Values represent the mean ± standard deviation of 3 independent experiments. Statistical analysis was performed using the unpaired *t*-test. *n.s*., without significance and *** *p*-value < 0.001.

**Figure 3 cancers-16-00005-f003:**
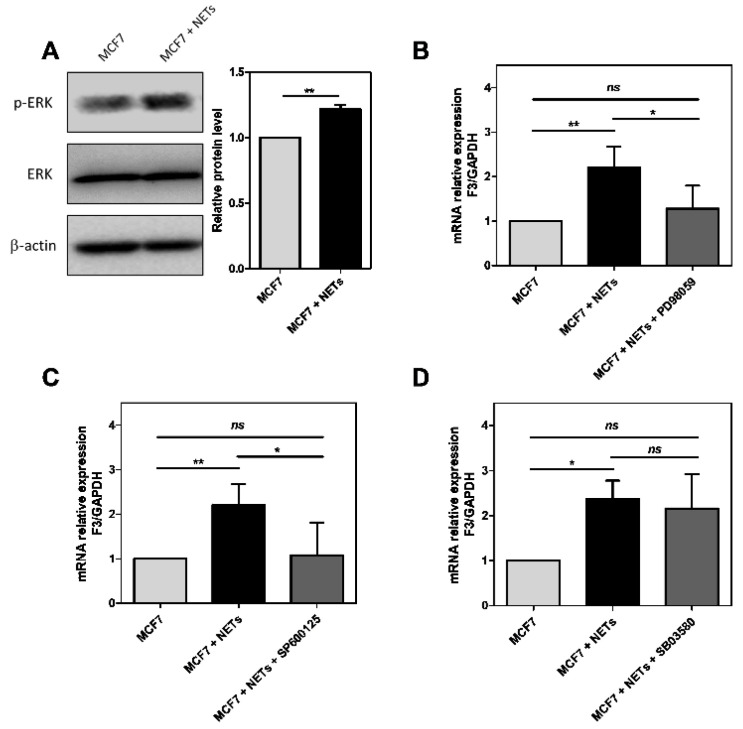
MAPK signaling pathways are required for NET-induced TF expression in MCF7 cells. Western blotting analysis of the p-ERK levels after 24 h of treatment with NETs (500 ng/mL) in MCF7 cells. Total ERK protein was used as the loading control. Densitometry was performed using Image J (**A**). MCF7 cells were previously incubated for 1 h with the following inhibitors: 50 µM PD98059 (**B**), 50 µM SP600125 (**C**), and 10 µM SB203580 (**D**). Then, they were incubated for 24 h with NETs (500 ng/mL), and TF mRNA expression was analyzed by qRT-PCR. *GAPDH* was used as an endogenous gene, and the ΔΔCt method was performed. Graphs represent the mean of three Western blotting and three qRT-PCR independent experiments ± standard deviation. Statistical analysis was performed in GraphPad Prisma using the unpaired *t*-test (**A**) or one-way ANOVA (**B**–**D**). * *p*-value < 0.05, ** *p*-value < 0.01, and *ns* = no significance.

**Figure 4 cancers-16-00005-f004:**
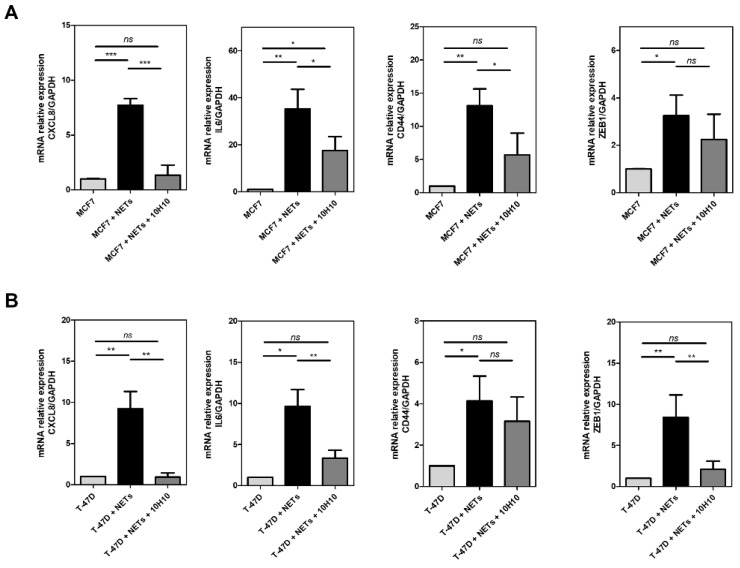
TF participates in the protumor response induced by NETs. MCF7 cells (**A**) and T-47D cells (**B**) were starved and further treated for 60 min with 10H10 antibody (50 µg/mL). Afterward, 500 ng/mL NETs were added for 24 h. Gene expression was evaluated by qRT-PCR using the ΔΔCT method. The analyzed genes were *CXCL8* (interleukin 8), *IL6* (interleukin 6), *CD44*, and *ZEB1*. *GAPDH* was used as the reference gene. Columns represent means ± SD of three independent experiments. Statistical analysis was performed using a one-way ANOVA test. * *p* < 0.05, ** *p* < 0.01, *** *p*-value < 0.001, and *ns* = no significance.

**Figure 5 cancers-16-00005-f005:**
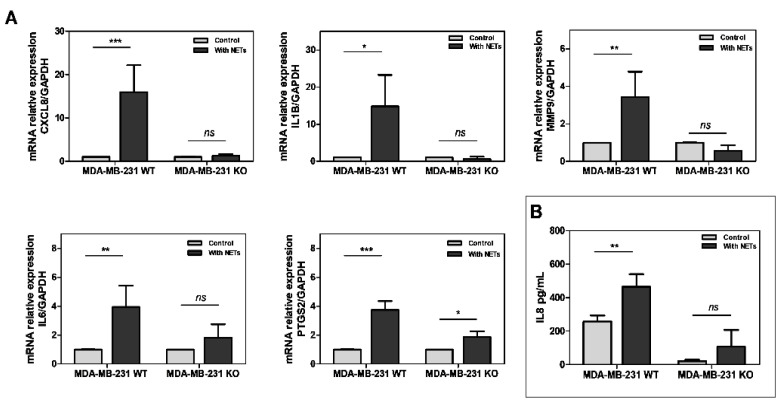
TF silencing suppresses NETs-driven protumor and proinflammatory responses in MDA-MB-231 cells. (**A**) MDA-MB-231 cells TF knockout (TF KO) or transfected with empty vector (TF WT) were treated with NETs (500 ng/mL) for 24 h and evaluated by qRT-PCR. The analyzed genes were *CXCL8* (interleukin 8), *IL1β* (interleukin 1β), *MMP9* (metalloproteinase 9), *IL6* (interleukin 6), and *PTGS2* (cyclooxygenase 2). GAPDH was used as an endogenous gene, and the ΔΔCt method was performed. (**B**) IL-8 antigen levels in the conditioned media of MDA-MB-231 TF WT or TF KO cells treated with NETs (500 ng/mL) for 24 h were determined using an enzyme-linked immunosorbent assay. Values represent the mean ± standard deviation of three independent experiments. * *p*-value < 0.05, ** *p*-value < 0.01, *** *p*-value < 0.001, and *ns* = no significance (unpaired *t*-test).

**Figure 6 cancers-16-00005-f006:**
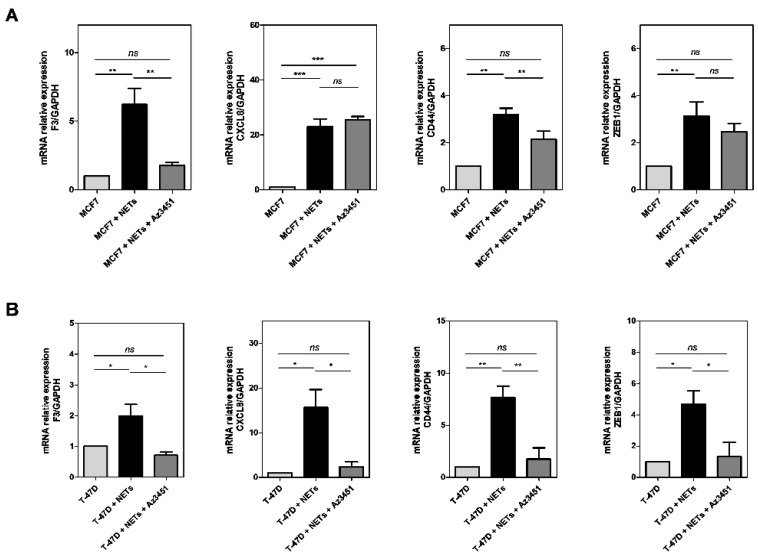
PAR2 participates in the NET-induced protumor response. MCF7 cells (**A**) and T-47D cells (**B**) were starved and further treated for 60 min with 10 µM Az3451. Afterward, 500 ng/mL NETs were added for 24 h. Gene expression was evaluated by qRT-PCR using the ΔΔCT method. The analyzed genes were *F3* (TF), *CXCL8* (interleukin 8), *CD44*, and *ZEB1*. *GAPDH* was used as a reference gene. Values represent the mean ± standard deviation of three independent experiments. * *p*-value < 0.05, ** *p*-value < 0.01, *** *p*-value < 0.001, and *ns* = no significance (one-way ANOVA).

**Figure 7 cancers-16-00005-f007:**
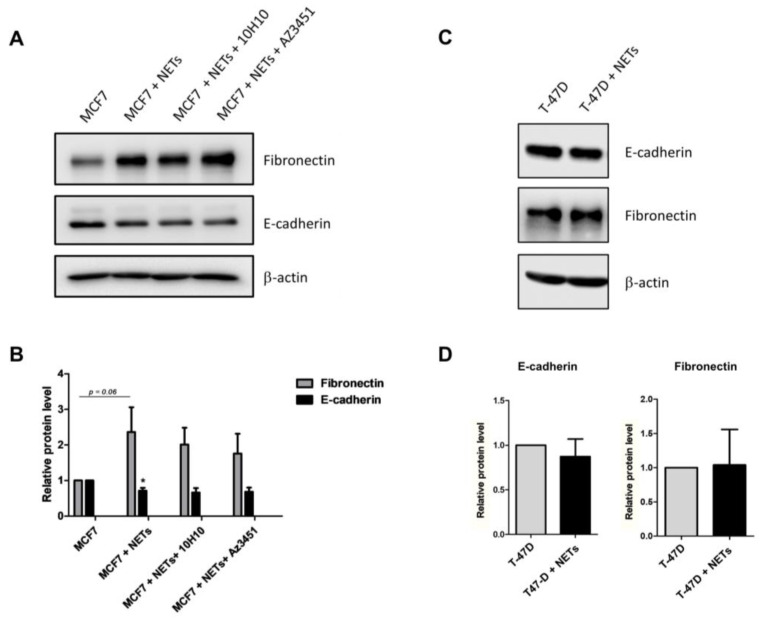
TF and PAR2 do not regulate the NET-driven EMT program. Western blotting analysis of the EMT markers (E-cadherin and fibronectin) in MCF7 (**A**) and T-47D (**C**) cells treated with Az3451 (10 µM) or 10H10 antibody (50 µg/mL) for 1 h, followed by stimulation with NETs (500 ng/mL) for 24 h. β-actin was used as the loading control. Representative image from three independent experiments. Densitometry was performed using Image J and graphs represent the mean ± standard deviation ((**B**)—MCF7 cells, (**D**)—T-47D cells). * *p*-value < 0.05 (one-way ANOVA).

**Figure 8 cancers-16-00005-f008:**
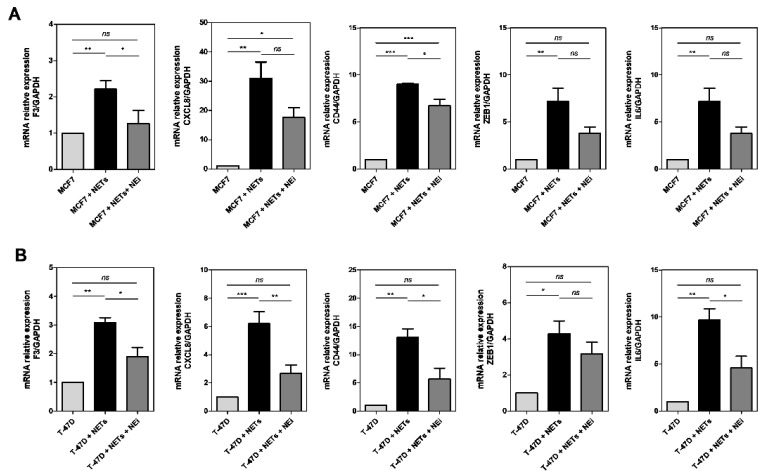
Inhibition of the neutrophil elastase impairs NET-triggered protumor response. MCF7 (**A**) and T-47D (**B**) cells were starved and treated with NEi (10 µM) for 1 h before NET treatment (500 ng/mL). After 24 h, samples were obtained to perform qRT-PCR analysis. The analyzed genes were *F3* (TF), *CXCL8* (interleukin 8), *CD44*, *ZEB1*, and *IL6* (interleukin 6). *GAPDH* was used as a reference gene. ΔΔCt method was performed. Values represent the mean ± standard deviation of three independent experiments. * *p*-value < 0.05, ** *p*-value < 0.01, *** *p*-value < 0.001, and ns = no significance (one-way ANOVA).

**Figure 9 cancers-16-00005-f009:**
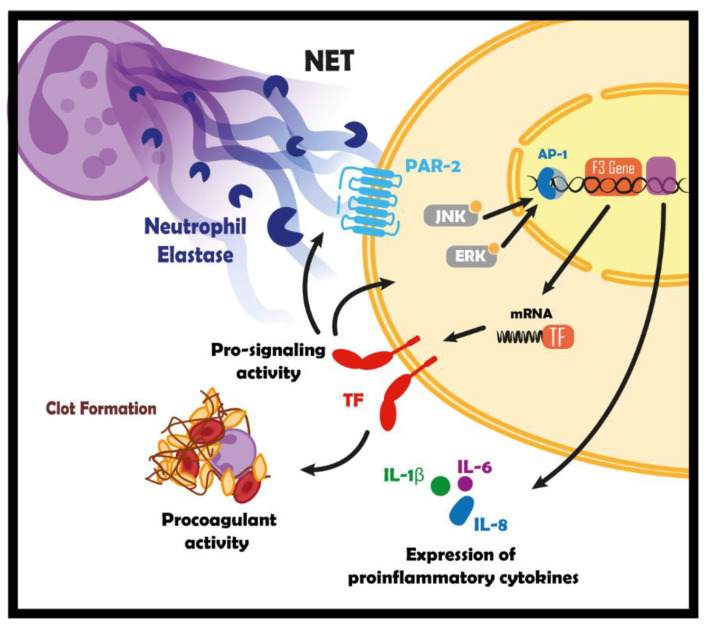
Schematic model connecting NETs and TF in tumor biology. Schematic representation of the NETs-triggered signaling in breast tumor cells through the interaction between components, such as neutrophil elastase, and PAR2, generating the expression of TF, and proinflammatory cytokines.

## Data Availability

The data presented in this study are available in this article (and Appendix A).

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
