# Peer review of "TF/PAR2 Signaling Axis Supports the Protumor Effect of Neutrophil Extracellular Traps (NETs) on Human Breast Cancer Cells"

_cancers, 2023, doi:10.3390/cancers16010005_

Round 1
Reviewer 1 Report
Comments and Suggestions for Authors
The Review “TF/PAR2 SIGNALING AXIS SUPPORTS THE PROTUMOR EFFECT OF NEUTROPHIL EXTRACELLULAR TRAPS (NETs) ON HUMAN BREAST CANCER CELLS” by Martins-Cardoso et al.
The Manuscript is addressing an important direction of immunosuppressive and cancerogenous role of tumor-associated neutrophils. The Manuscript by Martins-Cardoso et al. investigate the TF/PAR2 signaling axis impact on NETs-induced tumorogenesis. Authors analyzed influence of NETs on MCF7 and T47-D cell lines and find correlation between NETs treatment and TF or PAR2 gene expression. The Manuscript is well written and logically organized. The introduction includes detailed information describing the origin, gene expression, and role of tumor-resident neutrophils, and other molecular mechanisms of cancer progression. However, results consisting of stingy description of experimental design supplemented with broad speculations of NETs-tumor crosstalk. I found the Manuscript interesting and important, and suggesting authors to improve the figures and results (do not show only bars; bars should include individual values), expand supplementary data of some figures with raw data, such WB, mRNA expression profile or flow cytometry data.
My minor suggestions to the authors are the following:
1. The Results sections lacking any description of used databases together with performed in silico analysis. It is only showed as a final figure 1. Methods contains this information. I suggest duplicating it in Results.
2. NETs isolation description is also very short and refer to the Najmen et .al. Considering that authors used 500 nM of PMA for neutrophil induction and follow the ‘Najmen et .al.’ protocol – it may happen that residual amount of PMA can impact and stimulate MCF7 and T-47D cells. How authors avoid and control that this is not the case?
3. The MAPK signaling pathways analysis data represented in Figure 3 are not supported with raw WB blot data.
4. Supplementary Figure 1 does not have any figure capture.
5. Based on the different protein expression results authors jumping from one target to another in each section and figure. It would be beneficial for readers to include schematic expression of pathways targeted by inhibitors or agonists in figures.
Author Response
Responses are found in the attached PDF file.

Reviewer 2 Report
Comments and Suggestions for Authors
General: The paper describes the activation of tissue factor (TF) by neutrophils extracellular traps (NETs) that promote the pro-coagulant activity of breast cancer cells and their expression of pro-tumoral genes. The activation of TF is mediated by the ERK and JNK, but not p38 MAPKs. Once TF is expressed, it can bind to PAR2, activate it and induce some of the pro-tumoral genes. Below are my major concerns:
Major comments:
1. The authors do not present evidence of NETs formation in their protocol. An immunofluorescent image (at least in the supplementary data) should suffice.
2. The ability of TF to bind PAR1 is also known, but the authors focus exclusively on PAR2, without any explanation or a result negating PAR1, not even in silico analysis. This should at least be discussed.
3. Figure 1: Although significant, the r value of the correlation is low, raising doubts about the causality between TF or PAR2 and the NETs gene signatures. Please discuss.
4. Figure 3: The western blot shows only p-ERK, but total ERK (phosphorylated and non-phosphorylated) should be presented as well. The normalization should be p-ERK/total ERK.
5. Figure 4,5: the authors choose to look at IL-8, IL-6, CD44 (fig 4) or at IL-8, IL-1β, IL-6, MMP-9, COX2 (fig 5) at the mRNA levels only.
a. Please explain why these specific proteins, and not others, were selected.
b. Cytokines are regulated at multiple levels of regulation. Therefore, it is always more advisable to look at the final, secreted protein product. This was done for IL-8 (fig 5), but should be done for all secreted cytokines (IL-8, IL-6, IL1β, MMP-9).
c. The TME is immunosuppressive, and pro-inflammatory cytokines are found in very low levels. However, ERK and JNK that are activated by the TF/PAR2 axis could also activate anti-inflammatory cytokines, such as TGFβ. Those could be easily determined as well.
6. Figure 4, 6: The authors claim that NETs are linked to EMT activation. However, they only looked at Zeb1 mRNA expression, and not at other EMT inducers, such as Snail, Slug or Twist1, that might be alternatively activated.
7. Figure 7: the authors look at E-cadherin and fibronectin to determine that EMT is not significantly affected by NETs, although migration was affected by NETs and TF expression (Supp. data). As WB is an inaccurate, semi-quantitative method that cannot identify minor changes, I suggest that the WB data be supported with another method (e.g., immunofluorescence). In addition, the experiments were repeated only twice, with wide SD bars that question the ability to perform statistics on these data.
8. The authors are advised to add a schematic figure that describes the activated signaling. Which genes are activated by both TF and PAR2, and which by TF but not by PAR2? What is the order of events: do NETs activate ERK/JNK that induce TF expression that binds to PAR2 and activate the secretion of the pro-inflammatory proteins? Can NETs directly activate the secretion of the pro-inflammatory (and maybe anti-inflammatory) cytokines just by activating the MAPKs?
Author Response

(The authors gave the same response as above.)

Round 2
Reviewer 2 Report
Comments and Suggestions for Authors
The authors have addressed my concerns